# Hierarchical Nickel Cobalt Phosphide @ Carbon Nanofibers Composite Microspheres: Ultrahigh Energy Densities of Electrodes for Supercapacitors

**DOI:** 10.3390/nano13222927

**Published:** 2023-11-10

**Authors:** Jinqiao Zhang, Meiling Cen, Tao Wei, Qianyun Wang, Jing Xu

**Affiliations:** College of Materials and Metallurgy, Guizhou University, Guiyang 550025, China; gs.jqzhang21@gzu.edu.cn (J.Z.); gs.qywang22@gzu.edu.cn (Q.W.)

**Keywords:** supercapacitors, carbon nanofibers, CoNiP, electrochemistry performance

## Abstract

Supercapacitors (SCs) are widely used in energy storage devices due to their superior power density and long cycle lifetime. However, the limited energy densities of SCs hinder their industrial application to a great extent. In this study, we present a new combination of metallic phosphide–carbon composites, synthesized by directly carbonizing (Ni_1−x_Co_x_)_5_TiO_7_ nanowires via thermal chemical vapor deposition (TCVD) technology. The new method uses one-dimensional (1D) (Ni_1−x_Co_x_)TiO_7_ nanowires as precursors and supporters for the in situ growth of intertwined porous CNF microspheres. These 1D nanowires undergo microstructure transformation, resulting in the formation of CoNiP nanoparticles, which act as excellent interconnected catalytic nanoparticles for the growth of porous 3D CNF microspheres. Benefiting from the synergistic effect of a unique 1D/3D structure, the agglomeration of nanoparticles can effectively be prevented. The resulting CNF microspheres exhibit an interconnected conductive matrix and provide a large specific surface area with abundant ion/charge transport channels. Consequently, at a scanning rate of 10 mV s^−1^, its specific capacitance in 1.0 M Na_2_SO_4_ + 0.05 M Fe(CN)_6_^3−/4−^ aqueous solution is as high as 311.7 mF cm^−2^. Furthermore, the CoNiP@CNFs composite film-based symmetrical SCs show an ultrahigh energy density of 20.08 Wh kg^−1^ at a power density of 7.20 kW kg^−1^, along with outstanding cycling stability, with 87.2% capacity retention after 10,000 cycles in soluble redox electrolytes. This work provides a new strategy for designing and applying high-performance binary transition metal phosphide/carbon composites for next-generation energy storage devices.

## 1. Introduction

In recent years, the rapid development of renewable energies (e.g., solar, hydro, wind, etc.) as well as electric vehicles has led to a rising urgent demand for humans to find a green energy storage device [1,2,3]. Among the various existing energy storage devices, supercapacitors (SCs) represent one of the most promising energy storage technologies due to their high power density, ultralong cycling life span, and fast charge/discharge process [4,5,6]. Compared to batteries, their drawback of low energy density (3–5 Wh kg^−1^) has been a bottleneck problem that restricts their possible industrial applications [7].

SCs consist of anode and cathode electrodes, which are immersed in an aqueous or nonaqueous electrolyte and separated by separators that ensure the channels of electrons and ions. Generally, based on the energy storage mechanism, SCs can be divided into electronic double layer capacitors (EDLCs) and pseudo capacitors (PCs). The former obtain capacitance through the accumulation of electrostatic charges at the electrode/electrolyte interface. The latter depend on the occurrence of the rapid Faraday reaction on the surface of the electrode. Hence, it is an efficient method to develop an electrode with a high specific active surface for improving capacitance as well as energy density. Nanosized conductive carbon electrodes (e.g., carbon nanofibers (CNFs) [8], carbon nanotubes (CNTs) [9], activated carbon (AC) [10], and graphene [11,12]) are an ideal EDLC material due to their high conductivity, large surface area, and low weight. Taking CNFs as an example, they normally exhibit cylindrical or conical structures with lengths varied in the order of micron units, which can provide a highly accessible active surface and ion/charge distribution. At present, carbon material-based EDLC can only achieve very low energy densities of 3–5 Wh kg^−1^ [13]. Hence, the introduction of soluble redox-active species (e.g., ferrocene) into the aqueous electrolyte, or the coating of redox-active species (e.g., metal oxides) onto the capacitor electrodes are effective strategies. The resulting PCs exhibit much higher capacitance than EDLCs. For example, the capacitance of a CNF-based PC (constructed through the introduction of Fe(CN)_6_^3−/4−^ redox couples) is about four times higher than that of a CNF-based EDLC [14].

The chemical vapor deposition (CVD) technique is an efficient method for the synthesis of CNFs. Suitable substrate, carbon source, and catalyst are the important parameters in this technique [15,16]. For example, nano-scaled transition metal catalysts (e.g., Cu [17], Ni [18], Co [19,20], Fe [21,22]) are usually employed. The type, size, and distribution of applied transition metal nanoparticles determines the properties of CNFs. Unfortunately, the agglomeration of these catalytic nanoparticles on the substrate is hard to control, resulting in reduced surface area of CNFs. In order to further increase accessible surface area, 3D porous metal substrates (e.g., Ti [23], Ni [24,25], or Fe [26] foam) are often applied for the growth of CNFs. For example, the Ni foam/CNFs electrode exhibits a maximum specific capacitance of 134.3 F g^−1^ at a scanning rate of 5 mV s^−1^ [27]. However, the low volumetric energy densities, low packing densities, and high cost of these metal foams hinder their practical feasibility for the construction of SCs. Indeed, these drawbacks of current CNFs electrodes hinder the further development of CNF SCs.

We are thus interested in designing various kinds of CNFs, with various morphologies [5,28]. For high performance SCs, CNFs capacitor electrodes must be free from binders or current collectors. Otherwise, this will lead to poor conductivity and increase the complexity of SCs construction. For a new type of CNF capacitors, agglomeration of catalytic nanoparticles should be prevented, and expensive 3D metal foam should be replaced by cheap flat substrate.

Herein, binder-free hierarchical CoNiP@CNFs composite films were synthesized on a Ti substrate and applied as capacitor electrodes for the construction of high-performance SCs. Differing from the traditional methods, in this work, the intertwined porous CNF microspheres were grown through the in situ conversion of a 1D nanowire “forest” by using thermal chemical vapor deposition (TCVD) technology. Owing to its excellent electric conductivity, corrosion resistance, and high melting point [29], titanium (Ti) was chosen as the substrate material for the growth of the CoNiP@CNFs composite electrode. Transition bimetallic–carbon compounds have been exploited as high performance electrode materials for SCs. Compared with those transition metallic-based materials, metallic–carbon composite electrodes exhibit more advantages [30,31]. The multicomponent system consisting of transition metals and carbon offers compelling interactions with both materials. For example, CNFs can provide efficient conductive pathways to transport generated charges, and can resist volume expansion of transition redox materials during the charge–discharge process. As a result, they possess much higher energy densities and long term stability in various extremely harsh conditions. In the first part of this work, the characterization and growth mechanisms of CoNiP@CNFs composite film are discussed. Then, the electrochemical (EC) performances (e.g., capacitances, cyclic stability, and energy and power densities) of such composite electrodes are described. Finally, a symmetrical SC demonstrator based on CoNiP@CNFs electrodes is constructed for evaluating its potential applications as an energy storage device.

## 2. Materials and Methods

### 2.1. Synthesis and Characterization of the Composites

A schematic illustration of the synthesized CoNiP@CNFs nanocomposites on a Ti substrate is shown in Appendix A. Generally, the synthesis of a nanocomposite consists of three steps. The preparation of (Ni_1−x_Co_x_)_5_TiO_7_ nanowires was carried out using previously reported work [32]. Briefly, composite materials were generated on the surface of Ti substrate using the PEO method, followed by impregnation and annealing at 900 °C for 1 h. The integration of Co into the nanowires was achieved by adding cobalt acetate to the PEO electrolyte. Detailed information of applied electrolytes is listed in Appendix A. In order to grow CNFs, the as-prepared PEO sample was treated in a thermal chemical vapor deposition (TCVD) reactor (YG-120610, Shanghai Yuzhi Technology Co., Ltd., Shanghai, China), at a gas pressure of 75 torr and a temperature of 700 °C. Acetylene (C_2_H_2_) gas was used as a carbon source and as a reducing gas for the growth of CNFs. For comparison, the nanocomposites synthesized with the absence of Co (without the addition of cobalt acetate in PEO electrolyte) were also prepared and named as NiP@CNFs throughout the text.

The surface morphology of the synthesized nanocomposite film was obtained using Field-Emission Scanning Electron Microscopy (FESEM, Regulus 8230, Japan). Their superficial elemental compositions were investigated through X-ray photoelectron spectroscopy (XPS, Thermo scientific ESCALAB Xi+, USA) using Al Kα radiation of 200 W, and their effective elemental composition was analyzed using XPS PEAK 41 software. To determine the phase composition of the nanocomposites, X-ray diffraction (XRD, Ultima IV, Japan) measurements were performed in the 2θ range of 10–80°, with a step size of 0.02° min^−1^ using Cu Kα radiation. Raman spectroscopy (LabRAM Odyssey, HORIA, Japan) was applied to characterize the orderliness and the defect degree of the prepared carbon fiber composite material. A high-resolution transmission electron microscope (HRTEM, FEI TF20, USA) was used to analyze the crystal structure and morphology.

### 2.2. Electrochemical Measurements

The electrochemical measurements were conducted on an electrochemical workstation (CHI660EEA21378, Shanghai Chenhua Co., Ltd., Shanghai, China) using a standard three-electrode system, with Ag/AgCl (saturated 3 M KCl) and Pt wires (diameter of 1 mm) as the reference electrode and counter electrode, respectively. The cyclic voltammetry (CV) and galvanostatic charge/discharge (GCD) curves of CoNiP@CNFs composite capacitor electrodes were measured at various scanning rates and current densities in different electrolyte solutions. The geometric active area of the working electrode was fixed at 0.0314 cm^2^. For EC measurement, both EDLCs and PCs have been manufactured based on prepared nanocomposite electrodes. For the construction of EDLCs, the electrolyte used was a 1.0 M Na_2_SO_4_ aqueous solution, while a 0.05 M K_3_Fe(CN)_6_/K_4_Fe(CN)_6_-contained 1.0 M Na_2_SO_4_ aqueous electrolyte was used for the PCs. In addition, the cyclic service lifetime of such nanocomposite-based SC was evaluated at a current density of 40 mA cm^−2^.

In order to estimate the energy density and power density of the CoNiP@CNFs-based capacitor electrode, a two-electrode system was applied. A symmetrical SC device was constructed using a Nafion film (50 µm thickness, N-115, Dupont, Wilmington, DE, USA) as the separator. The specific capacitance (C, F cm^−2^), energy density (E, Wh kg^−1^), and power density (P, W kg^−1^) of these SCs were calculated according to the reported method [33]. Finally, a symmetrical SC demonstrator was further constructed using the as-prepared composite electrodes in the redox electrolyte to light up a light-emitting diode (LED) array. Meanwhile, the voltage–time curve of the SC demonstrator was recorded.

## 3. Results and Discussion

### 3.1. Characterization and Growth Mechanism of CoNiP@CNFs Composite Film

The morphology of (Ni_1−x_Co_x_)_5_TiO_7_ nanowires before and after the TCVD process was recorded using FESEM. In a representative SEM image (Figure 1a), a large number of straight (Ni_1−x_Co_x_)_5_TiO_7_ nanowires were found densely covering the sample surface, resulting in the formation of “nanowired forests”. The diameters of the obtained nanowires range from 76 to 167 nm. Surprisingly, these nanowires are directly grown at different angles (illustrated with green color) and form flower-like spheres (illustrated with red spot). It has been demonstrated that these nanowires are actually grown on a TiO_2_ layer, where P, Co, and Ni were ionized and incorporated during the PEO process [34]. Different from Ni_5_TiO_7_ nanowires (Appendix A), these Co-containing nanowires exhibit rough surfaces with small “spots” (inset in Figure 1a). This observation implies the successful incorporation of Co during the high energy discharged PEO treatment. This explanation can be supported by the related EDX spectrum, which displays the signals of Co, Ni, Ti, O, and P (Figure 1b). These results are consistent with previous work, which confirms the formation of (Ni_1−x_Co_x_)_5_TiO_7_ nanowires after the PEO treatment [32].

After TCVD treatment in C_2_H_2_ gas for 90 min, the flower-like spherical structures of the nanowires disappeared (Figure 1c). Instead, the interconnected CNFs network, in the form of microspheres (average diameter is about 75 nm), emerged on the sample surface. It was noted that the (Ni_1−x_Co_x_)_5_TiO_7_ nanowires were converted into interconnected nanoparticles in the initial stage (i.e., after 10 min) of TCVD treatment (Appendix A). As evidenced, the related enlarged SEM (inset in Figure 1c) reveals a bright nanoparticle located at the top/bottom of the fibers. The EDX spectrum of the nanoparticle (Point 2 in Figure 1d) mainly exhibits the signals of Ni, Co, and P, while that of a nanofiber predominately exhibits the signal of C (Point 3 in Figure 1d). According to the vapor–liquid–solid (VLS) growth mechanism, the CoNiP nanoparticle acts as a catalyst for the growth of CNFs [35]. It is believed that the carbon/hydrocarbon atoms derived from the pyrolysis of C_2_H_2_ dissolve and diffuse in the surface CoNiP binary catalyst at high temperatures, resulting in the growth of CNFs. Besides, the pressure of the introduced C_2_H_2_ gas in TCVD determines the morphology of nanofibers (Appendix A). For example, with the increase of C_2_H_2_ gas from 3.75 torr to 75 torr, the number and density of carbon nanofibers gradually increases and can even cover nanoparticles. The cross-sectional SEM image of CoNiP@CNFs (Figure 1e) reveals that it mainly consists of two layers: a fibrous top layer, and a compact, thick inner layer. The thickness of the interlayer is about 31.1 µm. A representative EDX line profile was applied to analyze the element distribution along this composite film (inset in Figure 1e). The Ti element content was found to be much higher in the inner layer than in the outer layer. In contrast, much higher content of the C element was found in the outer layer, owing to the existence of CNFs. This indicates that the outer part of the TiO_2_ phase has been carbonized in the C_2_H_2_ atmosphere during the TCVD treatment. Interestingly, the elements of Ni, Co, and P were mainly located beneath the fibrous outer layer; this is because the Ni–Co binary phosphide acts as an excellent catalyst and is eventually covered by CNFs during the TCVD treatment.

The XRD technique was used to study the phase, as well as the crystal structure, of the synthesized nanomaterials. The XRD spectra of CoNiP@CNFs nanocomposite films are shown in Figure 1f. A small, broad peak appeared at a low 2θ area from 10 to 20°, indicating the formation of amorphous CNFs. Furthermore, the diffraction peaks at 27.4°, 54.2°, and 36.0° are attributed to the characteristic reflections of the rutile TiO_2_ phase (JCPDS No. #21-1276), which is formed through PEO treatment of Ti substrate at high voltage [36]. The diffraction peaks at 36.0°, 41.8°, and 76.3° can be attributed to TiC (JCPDS No. #32-1383). Besides, diffraction peaks centered at 41.0°, 44.9° and 54.7° match well with CoNiP (JCPDS No. #71-2336), which exhibits excellent catalytic activity for C_2_H_2_ decom-position and growth of CNFs. The appearance of CoNiP matches well with the results of the EDS spectrum recorded through SEM imaging.

The morphology of the fabricated CoNiP@CNFs was further investigated through detailed TEM investigations. Low-magnification TEM imaging reveals a CNF with a diameter of about 154 nm (Figure 2a). Figure 2b,c were recorded at an interface of CoNiP and CNFs, suggesting that such materials exhibit multicomponent systems. The CNF exhibits a fibrous form and CoNiP displays a ball-like structure. As shown in Figure 2d, the HR-TEM image of the CoNiP nanoparticles displays the lattice fringes at about 0.205 nm, in agreement with (002) interplanar spacings of CoNiP [37]. The related SAED pattern (inset in Figure 2d) shows a clear polycrystalline diffraction ring, revealing the polycrystalline nature of the CoNiP. In comparison, the SAED pattern (inset in Figure 2e) recorded in the CNF region shows multiple diffraction rings. Hence, fabricated CNFs are mostly amorphous carbon. In addition, the EDS elemental mapping (Figure 2f) undoubtedly reveals the existence of constituent elements Ni, Co, and P in the interior region, while C is distributed around the entirety of CoNiP nanoparticles.

Raman spectroscopy was used to confirm the growth and carbonization degree of CNFs after the TCVD treatment of CoNiP@CNFs nanocomposites. The Raman spectra of CoNiP@CNFs shows two distinct peaks at 1350 cm^−1^ and 1580 cm^−1^, which is associated with defect-induced D and ordered G bands, respectively (Figure 3a). As a comparison, the Raman measurement of NiP@CNFs was also conducted (Appendix A). The R values of I_D_/I_G_ calculated from CoNiP@CNFs and NiP@CNFs are 2.43 and 1.35, respectively. The higher D/G intensity ratio of CoNiP@CNFs suggests a lower degree of order in the CNFs and better functionalization of CNFs, which is attributed to the high catalytic ability of interconnected CoNiP alloy nanoparticles towards carbonization [38].

In addition, X-ray photoelectron spectroscopy (XPS) was conducted to detect the specific composition and chemical bonding state of CoNiP@CNFs nanocomposites. In the XPS survey spectrum, the dominant element is carbon (at 83.12%), which indicates the successful carbonization of the catalytic nanowires and the growth of CNFs during the TCVD process (Appendix A). The high-resolution C 1s XPS of this nanocomposite material mainly shows four deconvoluted peaks (Figure 3b), which can be attributed to C-C (284.4 eV), C-O (285.6 eV), Ti-C (281.9 eV), and C=O (288.4 eV) binding energies [39,40]. In the high-resolution P 2p XPS spectrum (Figure 3c), the main peak is concentrated at the binding energy of 129.8 eV, demonstrating the metal phosphides [34], which is consistent with the CoNiP according to the XRD measurement. The peaks centered at Ti 2p_3/2_ (454.8 eV) and Ti 2p_1/2_ (460.5 eV) appear in the Ti 2p (Figure 3d) correspond to the Ti-C bond [39,41]. Regarding Ni 2p (Figure 3e), its peaks are mainly centered around 856.0 eV and 873.3 eV, which correspond to Ni 2p_3/2_ (Ni^II^) and Ni 2p_1/2_ (Ni^III^). The broad peaks at 862.0 eV and 880.0 eV are attributed to the oscillating satellite peaks of Ni 2p_3/2_ and Ni 2p_1/2_ [33,42]. In regard to the Co 2p XPS spectrum (Figure 3f), the binding energies of 780.1 and 795.2 eV correspond to Co^III^ (Co 2p_3/2_ and Co 2p_1/2_), while the characteristic peaks centered around 804.4 eV (satellite peak), 781.9, and 787.1 eV are attributed to Co^II^ and Co^IV^, respectively [43,44,45]. These results confirm the formation of CoNiP@CNFs nanoparticles. This nanocomposite material is expected to have electrochemical activity, due to the presence of active nickel and cobalt metal centers and conductive 3D network CNFs.

### 3.2. Supercapacitor Performance of CoNiP@CNFs Electrodes

The electrochemical performances of numerous hierarchical CoNiP@CNFs composite electrodes were tested in a three-electrode system. In order to choose a suitable electrolyte, firstly, the electrochemical properties of such composite electrodes were studied in three different electrolyte solutions (1.0 M H_2_SO_4_, 1.0 M KOH, and 1.0 M Na_2_SO_4_ aqueous solutions). As shown in Appendix A, CoNiP@CNFs composite electrode reveals the largest CV area in 1.0 M Na_2_SO_4_ aqueous solution; the estimated capacitances from the recorded CV curves are 26.4, 35.8, and 115.5 mF cm^−2^ at the same current density in 1.0 M H_2_SO_4_, 1.0 M KOH, and 1.0 M Na_2_SO_4_ aqueous solution, respectively. Therefore, 1.0 M Na_2_SO_4_ aqueous solution was used as an electrolyte for further EC tests of CoNiP@CNFs composite electrodes. The CV curves recorded at different scanning rates exhibit a rectangular shape within the potential window of −0.1~1.0 V (Figure 4a), indicating the ideal EDLC behavior of typical CNFs. At scanning rates of 10, 20, 50, and 100 mV s^−1^, the estimated capacitances are 226.7, 197.7, 158.3, and 124.5 mF cm^−2^, respectively. Compared to the CV curves of NiP@CNFs (Appendix A), those of CoNiP@CNFs were higher in voltammetric current, indicating larger available active sites for the diffusion of electrons and ions. The GCD curves of CoNiP@CNFs recorded at different current densities exhibit nearly perfect triangles (Figure 4c), suggesting good reversibility of these EDLCs. The estimated capacitances of the 1.0 M Na_2_SO_4_ electrolyte are 98.0, 116.7, 140.4, 170.3, and 216.0 mF cm^−2^ at current densities of 40, 30, 20, 10, and 5 mA cm^−2^, respectively. At the same current densities, the capacitances of NiP@CNFs are estimated at 3.5, 4.6, 6.0, 8.4, and 10.5 mF cm^−2^, respectively (Appendix A). Therefore, the capacitance of CoNiP@CNFs is about 20 times higher than that of NiP@CNFs, implying supreme EC storage ability. This can be attributed to the formation of interconnected dense CNFs, which exhibit extremely high specific surface area owing to the excellent catalytic properties of CoNiP nanoparticles. Additionally, the existence of the synergistic effects of binary metal (e.g., Co and Ni) phosphide species can enhance the EC performance of the composite electrode [46,47]. The charge storage mechanism consists of surface capacitive and diffusion-controlled contributions. The former derives from the fast ion adsorption/desorption on the electrode surface. The latter depends on the redox reaction. In order to estimate the Faradaic contribution and charge storage, the constant b value of the CoNiP@CNFs is calculated. According to power law [30,48,49], the calculated b value is ~0.76, indicating that both capacitive and diffusion processes occur on the CoNiP@CNFs composite electrode.

To further enhance the capacitive performance of CoNiP@CNFs composite electrodes, PCs were constructed by adding soluble redox species to aqueous electrolyte. During this process, 0.05 M K_3_Fe(CN)_6_/K_4_Fe(CN)_6_ redox species were dissolved in 1.0 M Na_2_SO_4_ solution. Overall, the related CV curves recorded in this redox electrolyte show a pair of redox peaks typical of PC materials at all scanning rates. Moreover, there are no significant changes in the CV curve shape of the CoNiP@CNFs electrode with the variation of scanning rate from 10 to 100 mV s^−1^, suggesting that the fabricated CoNiP@CNFs electrode is favorable for fast Faradaic reactions; and the electrode had a relatively low resistance [47]. Taking a 100 mV s^−1^ CV curve of CoNiP@CNFs as an example, the redox peaks of Fe(CN)_6_^3−/4−^ are located at about 0.43 and 0.11 V, revealing the presence of the redox reaction of the Fe(CN)_6_^3−/4−^ redox couple (Figure 4b). At scanning rates of 10, 20, 50, and 100 mV s^−1^, the estimated capacitances are 311.7, 277.7, 227.5, and 168.1 mF cm^−2^, respectively—approximately 1.5 times higher than the capacitances obtained from 1.0 M Na_2_SO_4_-based EDLCs. The GCD curves of CoNiP@CNFs-based PCs are shown in Figure 4d. The GCD curves are almost symmetrical, with obvious plateaus, which is consistent with their CV curves. This implies the high Coulombic efficiency and Faradaic characteristics of the fabricated composite electrodes. The corresponding specific capacitances at different current densities were calculated from the GCD curves. In 0.05 M Fe(CN)_6_^3−/4−^ + 1.0 M Na_2_SO_4_ solution, the calculated capacitances of a CoNiP@CNFs-based PC electrode are 308.3, 255.9, 210.6, 165.1, and 143.8 mF cm^−2^ at current densities of 5, 10, 20, 30 and 40 mA cm^−2^, respectively.

The Nyquist plots obtained using the EIS curve for CoNiP@CNFs were recorded in a 1.0 M Na_2_SO_4_ solution, and its charge transfer resistance (R_ct_) was estimated to be 3.9 Ω (Figure 4e), which is much lower than that of NiP@CNFs (R_ct_ 17.3 Ω in Appendix A). This indicates that Ni-Co binary phosphide might lead to higher electric conductivity.

### 3.3. Cyclic Stability and SC Mechanism

The cyclic stability of the CoNiP@CNFs composite electrode—an important parameter for its practical application—was also tested in a three-electrode system by using the GCD technique. The CoNiP@CNFs composite film-based EDLC and PC were constructed and conducted in 1.0 M Na_2_SO_4_ electrolyte and K_3_Fe(CN)_6_/K_4_Fe(CN)_6_-dissolved 1.0 M Na_2_SO_4_, respectively. In both solutions, this nanocomposite electrode exhibits generally stable capacitance retention (Figure 4f). The CoNiP@CNFs composite electrode retained 95.7% of its initial capacitance after 10,000 cyclic charge/discharges at a current density of 40 mA cm^−2^ in 1.0 M Na_2_SO_4_. For CoNiP@CNFs composite film-based PC, capacitance remains as high as 90.0% after 10,000 cycles. This indicates the high stability of the applied composite material, leading to its excellent capacitance retention. The surface of the CoNiP@CNFs composite electrode was then examined using SEM (Figure 5a). Minor structural change can be found after such a cyclic stability test. Raman spectroscopy (Figure 5b) was used to estimate the intensity ratio of the D and G bands. As expected, the intensity ratio I_D_/I_G_ is 2.35, obtained by Gaussian fitting; it is almost unchanged after the cyclic stability test. The XPS method was used to investigate the superficial chemical composition of CoNiP@CNFs after the cycling test. Note that, from the XPS spectrum of the composite electrode after the cycle tests, no obvious changes in the Co 2p or the Ni 2p were found after long charge/discharge cycles (Appendix A). However, differing from what is shown in Figure 3b, the characteristic peak of metal-C (at 281.9 eV) disappeared in the C 1s region (Figure 5c). Only a very small new peak (133.2 eV) appears in the P 2p region (Figure 5d), which is assigned to phosphate (P-O) [34]. This indicates that the surface oxidation reaction occurred on the CoNiP nanoparticles, which may explain the small diminishment of the capacitance performance after 10,000 cycles. The profile of CV (Figure 5e) remains almost unchanged before and after the stability test. An EIS was also performed to study the electrical conductivity change after the cyclic test (Figure 5f). The related R_ct_ after the cyclic stability test is only 6.0 Ω, increased very slightly after the cycling process. In order to investigate the role of CNFs in such composite materials, a cyclic test was conducted on a CoNiP electrode (namely without CNF) (Appendix A). Surprisingly, only about 29.2% capacitance remained. Hence, we believe that the presence of CNFs can successfully suppress the phase transition on the surface of CoNiP. Besides, the CNFs skeleton can substantially minimize the volumetric deformation of CoNiP during charging/discharging cycles [50].

### 3.4. Energy and Power Densities

Symmetrical EDLC and PC devices based on a two-electrode system were assembled using the CoNiP@CNFs composite electrode. The related SC properties were investigated (Figure 6a). In both the 1.0 M Na_2_SO_4_ and 0.05 M Fe(CN)_6_^3−/4−^ + 1.0 M Na_2_SO_4_ solution electrolytes, a potential window of 3.2 V can be realized in CV curves, indicating a large potential window. The GCD curves of CoNiP@CNFs in 1.0 M Na_2_SO_4_ and 1.0 M Na_2_SO_4_ + 0.05 M K_3_Fe(CN)_6_/K_4_Fe(CN)_6_ were measured at a current density of 70 mA cm^−2^, as displayed in Figure 6b. Consistent with the CV analysis above, the PC-based GCD curves reveal longer charge and discharge time. In addition, the GCD profiles in both systems present a small IR drop, indicating quick charge transmission for CoNiP@CNFs. The GCD curves of CoNiP@CNFs in EDLC and PC solutions for various current densities, varying from 80 to 200 mA cm^−2^, are shown in Appendix A. Moreover, the CoNiP@CNFs composite electrode also displays a good life span. Even after 10,000 cyclic charge/discharges, over 94.8% and 87.2% of the initial capacitance were retained in the CoNiP@CNFs-based EDLCs and PCs, respectively (Figure 6c), consistent with cyclic retention in a three-electrode system. The coulombic efficiencies were further calculated according to the literature [30,51]. The CoNiP@CNFs maintain marvelous coulombic efficiency along the cyclic charge/discharge as EDLCs (~99.6%) and PCs (~101.2%), respectively (Figure 6c).

The energy density and power density of the CoNiP@CNFs composite film-based symmetrical SC device were calculated using a two-electrode system also used for 1.0 M Na_2_SO_4_ and 0.05 M K_3_Fe(CN)_6_/K_4_Fe(CN)_6_ + 1.0 M Na_2_SO_4_ electrolytes. The calculated energy and power densities of CoNiP@CNFs- and NiP@CNFs-based EDLCs and PCs are summarized in the Ragone diagram. Generally, CoNiP@CNFs-based SCs exhibit much higher energy and power density than that of NiP@CNFs. As shown in Figure 6d, the CoNiP@CNFs-based EDLC constructed in a 1.0 M Na_2_SO_4_ electrolyte provides a maximum energy density of 7.15 Wh kg^−1^ and a power density of 3156.8 W kg^−1^, while NiP@CNFs-based EDLC can only supply a maximum energy density of 4.35 Wh kg^−1^ and a power density of 312.0 W kg^−1^ (Appendix A). Clearly, the CoNiP@CNFs composite film-based PCs have higher energy and power densities than the CoNiP@CNFs-based EDLCs. For the CoNiP@CNFs-based PC device, a maximum energy density of 20.08 Wh kg^−1^ is achieved, together with a power density of 7.20 kW kg^−1^. Such high energy density of CoNiP@CNFs-based SCs surpasses most available traditional capacitors, as well as other SCs, and can even compete with Li-batteries. The high EC performance can be attributed to the hierarchical structural characteristics of CoNiP@CNFs composites, which exhibit unique microstructures of interconnected CNF-formed microspheres with extremely high specific active surface areas. In addition, the CoNiP binary metal phosphide, delivered from the incorporation of Co with Ni through PEO technology, provides enhanced electric conductivity, which accelerates the ion/electron transport.

To evaluate the potential application of CoNiP@CNFs composite film-based PCs, a symmetrical SC demonstrator was constructed by using 0.05 M K_3_Fe(CN)_6_/K_4_Fe(CN)_6_ + 1.0 M Na_2_SO_4_ redox electrolyte. As shown in Figure 7a, an array of 19 red LEDs were lit up with a working voltage of 1.8–2.0 V. The instantaneous operating voltage of this SC demonstrator was measured at 2.57 V with a voltammeter (inset in Figure 7a). The voltage–time curve of such an energy storage device was also recorded (Figure 7b). The LED array lit for 3 s with a charging time of 2.8 s. After eight cyclic charge/discharges, a minor change in the voltage–time curve shape can be observed. Therefore, the CoNiP@CNFs composite film-based electrode is a suitable candidate for the construction of high performance SC capacitor electrodes, and has considerable potential for application as an energy storage device in the future.

## 4. Conclusions

A novel, binder-free, hierarchical CoNiP@CNFs composite film has been successfully synthesized on a Ti substrate. Owing to the unique characteristics of PEO treatment, the obtained (Ni_1−x_Co_x_)TiO_7_ has nanowires with various angles that densely cover the surface, forming a “nanowire forest”. The interconnected CoNiP nanoparticles were formed in situ through the transformation of P contained in the (Ni_1−x_Co_x_)TiO_7_ “nanowired forest” during the TCVD treatment. Meanwhile, the TiO_2_ layer is carbonized to conductive TiC in C_2_H_2_ at high temperatures. Such a technique overcomes the agglomeration catalytic nanoparticles, replaces traditional metal foams, and obtains intertwined porous CNF microspheres, ensuring an extremely high specific surface area and enhancing channels for ion/charge transferability. Moreover, the synergistic effect of binary metal phosphide and CNFs can give electrodes a high specific capacity and low resistance. Together with soluble redox electrolytes, CoNiP@CNFs composite film-based PCs have been constructed, providing stable capacitance, high power densities, rapid charging/discharging rates, and in particular, extremely high energy densities. Therefore, this novel strategy for the fabrication of CoNiP@CNFs composite film will be highly useful for constructing high performance, binder-free, carbon material-based SCs, paving a reasonable path to the design of next-generation SC systems in aqueous solutions.

## Figures and Tables

**Figure 1 nanomaterials-13-02927-f001:**
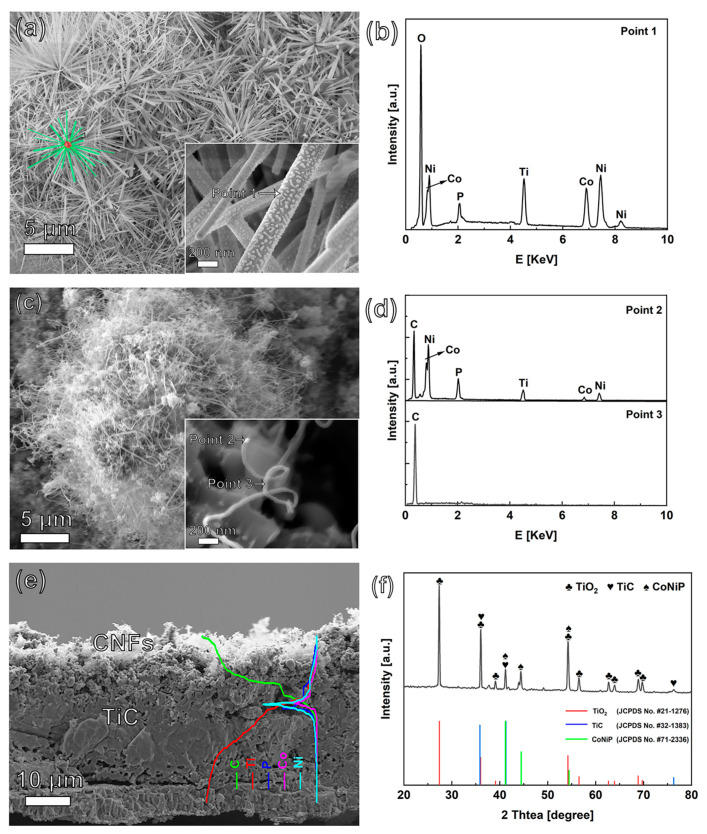
SEM images of P-contained (Ni_1−x_Co_x_)_5_TiO_7_ nanowires (**a**) before TCVD treatment, (**b**) at the related EDX spectra of “point 1”, (**c**) after TCVD treatment, and (**d**) at their corresponding EDX spectra of “point 2” and “point 3”; (**e**) SEM cross-sectional image of CoNiP@CNFs composite film, and (**f**) XRD spectra of CoNiP@CNFs nanocomposite. The insets in figures (**a**,**c**) show their related enlarged SEM images. The inset in figure (**e**) is one EDX line profile.

**Figure 2 nanomaterials-13-02927-f002:**
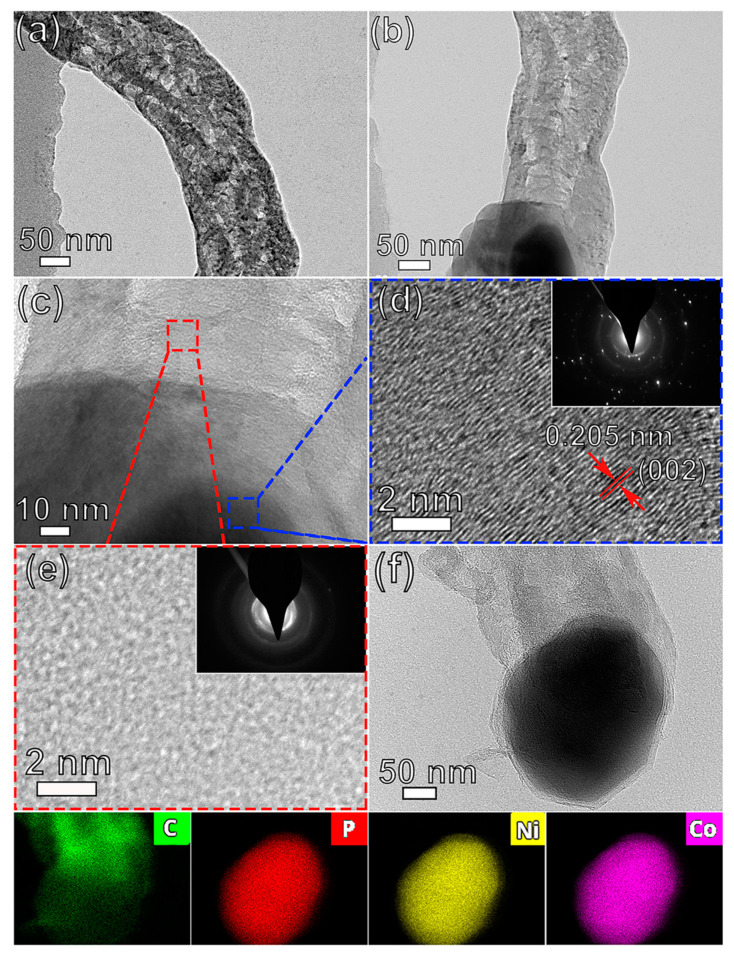
TEM images of (**a**) a CNF and (**b**) a CoNiP@CNFs nanocomposite, and (**c**–**e**) HR-TEM images of (**b**), a CNF, and a CoNiP nanoparticle, respectively. The insets in figures (**d**,**e**) show their SAED patterns. (**f**) HAADF-STEM image and corresponding EDS maps of C, P, Ni, and Co taken from an individual CoNiP@CNFs nanocomposite.

**Figure 3 nanomaterials-13-02927-f003:**
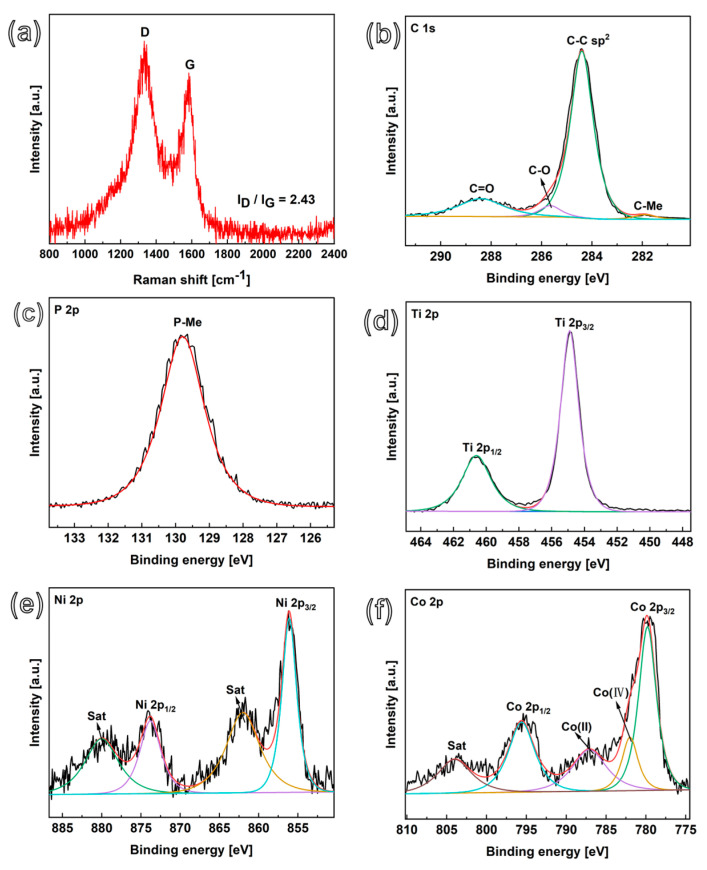
Raman spectra (**a**) and deconvoluted XPS spectra of (**b**) C 1s, (**c**) P 2p, (**d**) Ti 2p, (**e**) Ni 2p, and (**f**) Co 2p core levels of CoNiP@CNFs nanocomposites.

**Figure 4 nanomaterials-13-02927-f004:**
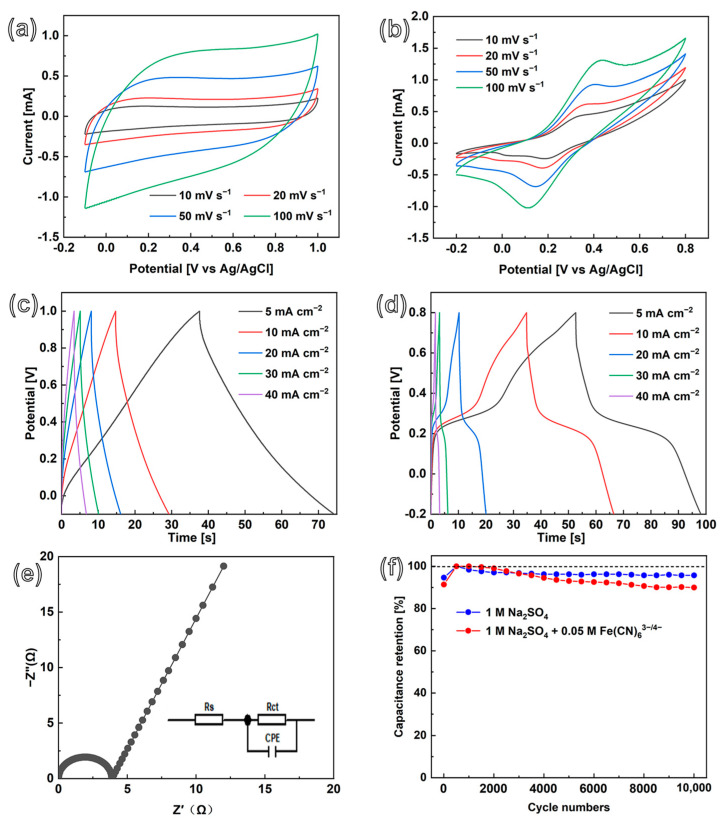
(**a**,**b**) CV and (**c**,**d**) GCD curves of CoNiP@CNFs composite capacitance electrodes in (**a**,**c**) 1.0 M Na_2_SO_4_ and (**b**,**d**) 0.05 M Fe(CN)_6_^3−/4−^ + 1.0 M Na_2_SO_4_ solution. (**e**) EIS curve in 1.0 M Na_2_SO_4_ solution and (**f**) capacitance retentions in 1.0 M Na_2_SO_4_/1.0 M Na_2_SO_4_ + 0.05 M Fe(CN)_6_^3−/4−^ solution.

**Figure 5 nanomaterials-13-02927-f005:**
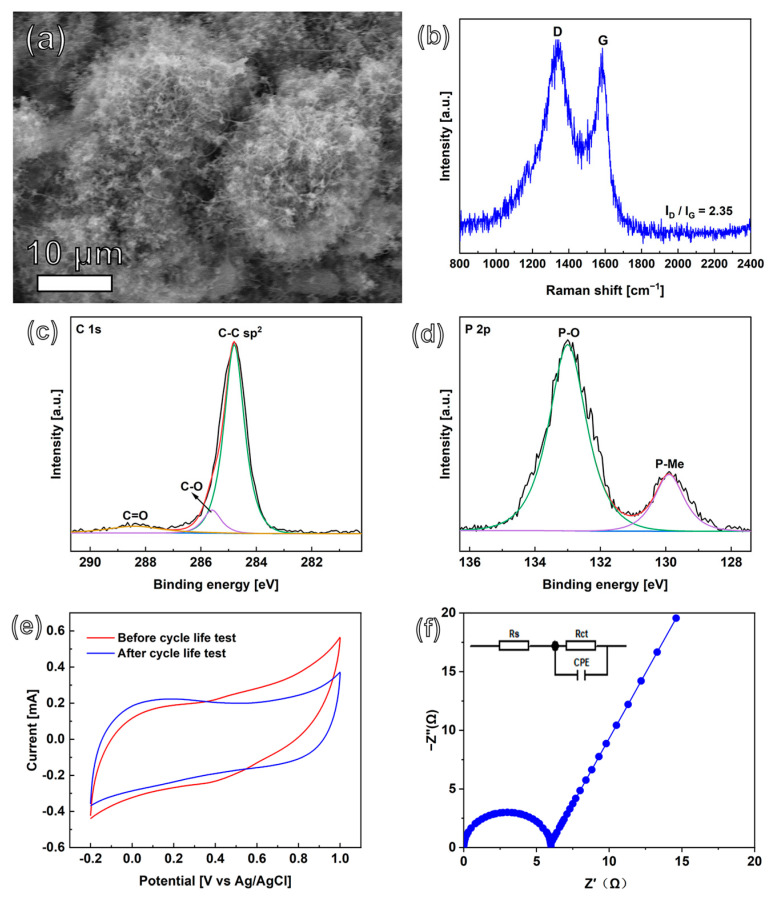
(**a**) SEM image, (**b**) Raman spectra, XPS spectra of (**c**) C 1s, (**d**) P 2p core levels, (**e**) CV curves in 1.0 M Na_2_SO_4_ solution, and (**f**) EIS curve in 1.0 M Na_2_SO_4_ solution of CoNiP@CNFs composite after cycle life test.

**Figure 6 nanomaterials-13-02927-f006:**
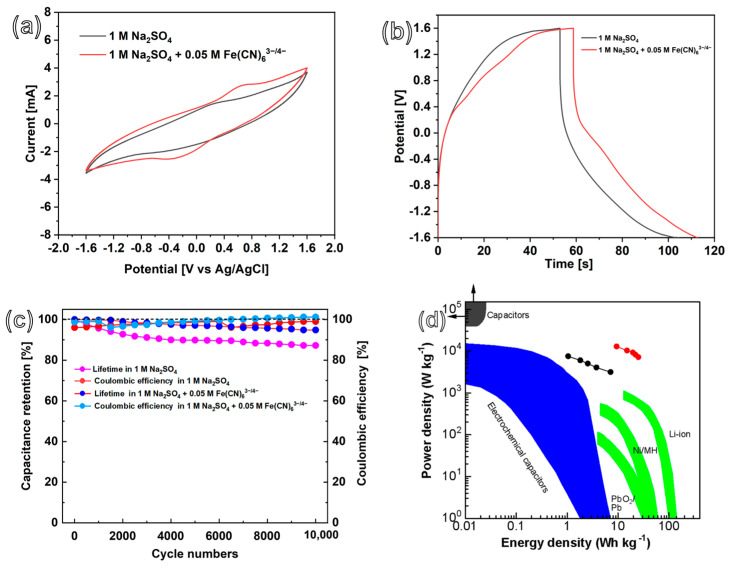
(**a**) CV, (**b**) GCD curves, and (**c**) capacitance retentions and coulombic efficiencies of CoNiP@CNFs composite capacitance electrodes, measured in 1.0 M Na_2_SO_4_ and 0.05 M Fe(CN)_6_^3−/4−^ + 1.0 M Na_2_SO_4_ solution with a two-electrode system. And (**d**), a comparison of Ragone plots of the CoNiP@CNFs-based EDLCs in 1.0 M Na_2_SO_4_ (black dotted line) and PCs in 0.05 M Fe(CN)_6_^3−/4−^ + 1.0 Na_2_SO_4_ (red dotted line) with those of traditional capacitors, ECs, and batteries. Reproduced with permission [52]. Copyright 2008, Nature Publisher.

**Figure 7 nanomaterials-13-02927-f007:**
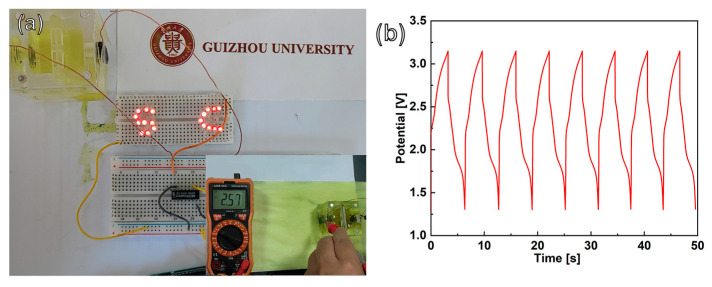
(**a**) Photograph of a SC demonstrator using a pair of CoNiP@CNFs composite capacitor electrodes (illuminated “SC”) and (**b**) voltage–time curve of this stand-alone PC demonstrator. The inset in Figure 7a shows its instantaneous operating voltage.

## Data Availability

Data are contained within the article and Appendix A.

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
