# Peer review of "Hierarchical Nickel Cobalt Phosphide @ Carbon Nanofibers Composite Microspheres: Ultrahigh Energy Densities of Electrodes for Supercapacitors"

_nanomaterials, 2023, doi:10.3390/nano13222927_

Round 1

Reviewer 1 Report

Comments and Suggestions for Authors

Hierarchical Nickel Cobalt Phosphide @ Carbon Nanofibers Composite Microspheres: Ultrahigh Energy Densities of Electrodes for Supercapacitor

This work presents the combination of metal phosphide-carbon composites synthesized by directly carbonizing nanowires via thermal chemical vapor deposition (TCVD) technology. CoNiP@CNFs composite film based symmetric SCs reveals an ultrahigh energy density. Come results are interesting, however, following revisions should be made before publication:

1.      The abstract section is prolix and unable to highlight the significant finding and objectives. More quantitative information’s should be highlighted.

2.      The rational design of metal-carbon interaction can be explained in the introduction section with the following articles:

doi.org/10.1016/j.compositesb.2022.110339, doi.org/10.1016/j.est.2023.106713

3.      JCPDS file should be indexed in XRD patterns. In figure 2 c, p 2p fitting is not proper.

4.      The electrochemical activities of all samples should be compared in same scan rates and current densities, should be presented in main manuscript. Coulombic efficiencies also should be explained.

5.      Information related to symmetric supercapacitors, including CV, GCD and stability is necessary. Pictorial information for the cell output can be demonstrated with voltammeter.

6.      The capacitive and faradaic contribution can be studied with the reference of following articles: doi.org/10.1016/j.cej.2021.132345

Author Response

Thank you very much for taking the time to review this manuscript. 

Please find the detailed responses in the attachment. The corresponding revisions/corrections highlighted with red color changes in the re-submitted files.

Reviewer 2 Report

Comments and Suggestions for Authors

The author describes the “Hierarchical Nickel Cobalt Phosphide@Carbon Nanofibers Composite Microspheres: Ultrahigh Energy Densities of Electrodes for Supercapacitor”. This original article is quite interesting from a technological point of view. The author should revise their manuscript based on the comments and suggestions. I recommended a Major revision of the manuscript. 

The Major suggestions are below:

  1. In the abstract, the author can state a clear research question to convey the main objective of this research work.
  2. The author should add more related work and discussion in the introduction section.
  3. Why author use Ti substrate for CoNiP@CNFs preparation? Is there any specific reason?
  4. The SEM images are unclear, so the author should provide quality SEM images.
  5. The author should provide the TEM images of the prepared samples.
  6. The authors should thoroughly review the manuscript for spelling and grammar errors. Numerous mistakes have been identified, and it is essential to address them diligently.
  7. A complete understanding of the SC mechanism, post-morphology, and compositional changes can be studied with XPS and Raman spectra analysis for the prepared material.
  8. The author should provide the prepared materials' after stability CV and EIS spectra.

Comments on the Quality of English Language

 Minor editing of the English language is required

Author Response

Dear Reviewer,

Thank you very much for taking the time to review this manuscript.

Please find the detailed responses in the attachment. The corresponding revisions/corrections highlighted with red color changes in the re-submitted files.

Round 2

Reviewer 1 Report

Comments and Suggestions for Authors

Accept